# Isoprene-Degrading Bacteria from Soils Associated with Tropical Economic Crops and Framework Forest Trees

**DOI:** 10.3390/microorganisms9051024

**Published:** 2021-05-10

**Authors:** Toungporn Uttarotai, Boyd A. McKew, Farid Benyahia, J. Colin Murrell, Wuttichai Mhuantong, Sunanta Wangkarn, Thararat Chitov, Sakunnee Bovonsombut, Terry J. McGenity

**Affiliations:** 1Department of Biology, Faculty of Science, Chiang Mai University, Chiang Mai 50200, Thailand; toungporn_u@cmu.ac.th (T.U.); tara.chitov@gmail.com (T.C.); 2Graduate School, Chiang Mai University, Chiang Mai 50200, Thailand; 3School of Life Sciences, University of Essex, Colchester CO4 3SQ, UK; boyd.mckew@essex.ac.uk (B.A.M.); fbeny@essex.ac.uk (F.B.); 4School of Environmental Sciences, Norwich Research Park, University of East Anglia, Norwich NR4 7TJ, UK; J.C.Murrell@uea.ac.uk; 5Enzyme Technology Research Team, Biorefinery and Bioproduct Technology Research Group, National Center for Genetic Engineering and Biotechnology, Pathumthani 12120, Thailand; wuttichai.mhu@biotec.or.th; 6Department of Chemistry, Faculty of Science, Chiang Mai University, Chiang Mai 50200, Thailand; sunanta@chiangmai.ac.th; 7Environmental Science Research Center (ESRC), Chiang Mai University, Chiang Mai 50200, Thailand

**Keywords:** isoprene, volatile hydrocarbon, isoprene degradation, bacterial communities, isoprene-degrading bacteria, tropical soils, economic crops, framework forest trees

## Abstract

Isoprene, a volatile hydrocarbon emitted largely by plants, plays an important role in regulating the climate in diverse ways, such as reacting with free radicals in the atmosphere to produce greenhouse gases and pollutants. Isoprene is both deposited and formed in soil, where it can be consumed by some soil microbes, although much remains to be understood about isoprene consumption in tropical soils. In this study, isoprene-degrading bacteria from soils associated with tropical plants were investigated by cultivation and cultivation-independent approaches. Soil samples were taken from beneath selected framework forest trees and economic crops at different seasons, and isoprene degradation in soil microcosms was measured after 96 h of incubation. Isoprene losses were 4–31% and 15–52% in soils subjected to a lower (7.2 × 10^5^ ppbv) and a higher (7.2 × 10^6^ ppbv) concentration of isoprene, respectively. Sequencing of 16S rRNA genes revealed that bacterial communities in soil varied significantly across plant categories (framework trees versus economic crops) and the presence of isoprene, but not with isoprene concentration or season. Eight isoprene-degrading bacterial strains were isolated from the soils and, among these, four belong to the genera *Ochrobactrum*, *Friedmanniella*, *Isoptericola* and *Cellulosimicrobium*, which have not been previously shown to degrade isoprene.

## 1. Introduction

Isoprene (C_5_H_8_) is the second most abundant natural hydrocarbon in the atmosphere after methane [1]. It is estimated that approximately 500 million tonnes of isoprene are emitted into the atmosphere annually, mainly from tropical forests [2]. Isoprene is a highly reactive volatile organic compound, which can bond with free radicals and react with nitrogen oxides to, inter alia, form ozone, which is a powerful greenhouse gas [3]. Tropospheric ozone can also have an adverse effect on the health of humans and animals, especially on the respiratory system, and is potentially harmful to all other organisms, biodiversity and ecosystems [4].

Isoprene is mainly emitted by terrestrial plants. Some trees such as oak, poplar and eucalyptus are major producers of isoprene [2,5], with reported emissions of 100 (*Populus* spp.) and 50 (*Eucalyptus* spp.) μg g^−1^ (dry leaves) h^−1^ [5,6]. The global intensification of agriculture is likely to alter the isoprene balance in the local, regional and potentially global atmosphere. In many countries, such as Thailand, traditional small-scale farming is being replaced by large-scale crop plantation. Moreover, deforestation, which goes hand-in-hand with agricultural intensification, is a major environmental threat. Many strategies and methods have been employed to restore forest ecosystems. One approach to forest restoration that has proven very successful is the “Framework Species method” [7,8,9]. In the tropics, this method involves planting mixtures of approximately 20 pioneer and climax tropical forest tree species such as *Spondias axillaris*, *Prunus cerasoides* (wild Himalayan cherry), and *Ficus altissima* (council tree) [8]. Framework trees can improve forest productivity and nutrient cycles, and can attract and bring back wildlife so that the forest ecosystems can be restored.

The agricultural and forestry activities outlined above are intended to be beneficial to the food supply and for sustaining the environment. However, large amounts of isoprene are emitted by many framework forest tree species, such as council tree (*Ficus altissima*) [10], as well as from economic crops, such as oil palm (*Elaeis guineensis*) and rubber trees (*Hevea brasiliensis*) [10,11,12]. Soil is a sink for some of this isoprene [13,14]. There is also the potential for internal cycling (both production and consumption) of isoprene within soils [15], with a range of microbes being responsible for isoprene consumption, such as members of the genera *Arthrobacter*, *Nocardia, Nocardiodes, Rhodococcus, Bacillus, Alcaligenes, Ramlibacter, Variovorax, Klebsiella, Pantoea* and *Pseudomonas* [16]. Knowing the potential and the capacity of soil bacteria and bacterial communities to consume isoprene will lead to a better understanding of isoprene degradation mechanisms in soil and the global atmospheric isoprene budget. However, most of the research into isoprene consumption by bacteria has been conducted in temperate zones [14,17,18,19]. There is limited understanding of isoprene consumption by soil bacteria in the tropics, i.e., those that are associated with tropical plants and, to our knowledge, only a few studies have been reported [20,21].

Our aims were: (1) to determine the potential for isoprene degradation in tropical soils associated with different plants that are important in agriculture and forestry; (2) to identify changes in bacterial community composition during isoprene degradation; (3) to isolate and characterise soil bacteria responsible for isoprene degradation.

## 2. Materials and Methods

### 2.1. Soil Sampling

Soil samples (50 g) beneath selected framework forest tree species were collected from the top 3 cm of soil using autoclaved spoons and sterile bags, during three different seasons: the rainy season (September, 2017), winter season (January, 2018) and summer season (May, 2018). The samples were collected from a restored forest area in Doi Suthep-Pui National Park, Chiang Mai, Thailand (18°51′51″ N, 98°50′51″ E). For soil associated with economic crops, samples were collected from various locations in Kamphaeng Phet province, Northern Thailand (16°29′0.38″ N, 99°31′17.51″ E). Soil samples associated with each species (Table 1) were taken from beneath the canopies of the trees from five different locations to give five replicates for each species. Soil physicochemical features are presented in Appendix A.

### 2.2. Preparation of Soil Samples for Isoprene Degradation Measurement

Each soil sample was divided into two 1 g portions, which were incubated with different concentrations of isoprene. Each sample was cultured in 9 mL of minimal medium (pH 7.2) modified from Fahy et al. [22], and contained (per litre): 0.5 g NaCl, 0.5 g MgSO_4_·7H_2_O, 0.1 g CaCl_2_·2H_2_O, 1 g NH_4_NO_3_, 1.1 g Na_2_HPO_4_, 0.25 g KH_2_PO_4_, 50 mg Cycloheximide, 10 mg FeSO_4_·7H_2_O, 0.64 mg Na_2_EDTA.3H_2_O, 0.1 mg ZnCl_2_, 0.015 mg H_3_BO_3_, 0.175 mg CoCl_2_·6H_2_O, 0.15 mg Na_2_MoO_4_·2H_2_O, 0.02 mg MnCl_2_·4H_2_O, 0.01 mg NiCl_2_·6H_2_O, 0.05 mg *p*-Aminobenzoic acid, 0.02 mg Folic acid, 0.02 mg Biotin, 0.05 mg Nicotinic acid, 0.05 mg Calcium pantothenate, 0.05 mg Riboflavin, 0.05 mg Thiamine HCl, 0.1 mg Pyridoxine HCl, 0.001 mg Cyanocobalamin, and 0.05 mg Thioctic acid, in a 125 mL crimp-top serum vial. Phosphate salts were autoclaved separately from other salts, and the vitamins were filter-sterilised. The vial was sealed with a PTFE/silicone septum, and injected with different amounts of isoprene (7.2 × 10^6^ ppbv and 7.2 × 10^5^ ppbv) as the sole carbon source. Isoprene stocks were made as described by Acuña Alvarez et al. [23], with 10 mL of 99% isoprene (Sigma-Aldrich, St. Louis, MO, USA) in a 125 mL sterilised serum vial, crimp-sealed with a silicone-PTFE septum, and warmed to 30 °C for 15 min before transferring the isoprene in the headspace. For 7.2 × 10^6^ ppbv of isoprene, 1 mL of headspace isoprene gas was transferred to the enrichment medium and for 7.2 × 10^5^ ppbv of isoprene, 0.1 mL was used. Five microcosms of autoclaved soil were also included in each batch to differentiate isoprene degradation by microbial activity from other forms of loss. The microcosms were incubated at 27 °C for 96 h.

### 2.3. Analysis of Isoprene Degradation by Gas Chromatography-Flame Ionisation Detection (GC-FID)

To monitor residual isoprene concentrations in sealed vials, 100 μL of headspace gas was taken through the septum and directly injected into the sample port of an Agilent 6890N GC-FID equipped with a DB-1 J&W column (30 m long, 0.25 mm internal diameter and 0.25 µm stationary phase thickness), with injector temperatures of 250 °C, a column temperature of 50 °C and detector temperatures of 275 °C. Helium was used as the carrier gas at a constant flow of 1.4 mL min^−1^. Significance testing was performed using an ANOVA test with Jamovi software version 1.2 [24].

### 2.4. Isolation and Identification of Isoprene-Degrading Bacteria

To isolate isoprene-degrading bacteria, the enrichments (soil samples incubated with isoprene) were taken from the vials after 96 h incubation. A 1 mL portion of the enriched samples was diluted and spread on minimal medium agar and incubated for 21 days in 1 L glass desiccator containing 1 mL of 99% isoprene (Sigma-Aldrich, St. Louis, MO, USA). Representative colonies were picked and re-streaked until pure cultures were obtained. For species identification, DNA from the pure cultures was extracted according to the method described by Griffiths et al. (2000) [25]. The 16S rRNA gene was amplified using the primers 27F (5′-AGAGTTTGATCMTGGCTCAG-3′) and 1492R (5′-CGGTTACCTTGTTACGACTT-3′) [26] at the final concentration of 0.4 μM. AppTaq RedMix (2×; Appleton Woods Ltd., Birmingham, UK) was used in the PCR. The PCR conditions were: denaturing at 95 °C for 3 min, followed by 30 cycles of denaturing at 95 °C for 15 s, annealing at 55 °C for 15 s, extension at 72 °C for 30 s, and a final extension at 72 °C for 5 min. Amplicons were purified using a GenElute PCR Clean-Up Kit (Sigma, St. Louis, MO, USA). Sanger sequencing was performed using the above primers by Eurofins Genomics (Ebersberg, Germany). Low-quality bases were trimmed using Chromas software version 2.6.6. FASTA sequences were concatenated in Bioedit software version 7.0.5.3 [27]. The obtained sequences were deposited to the NCBI database (accession no. MW509981–MW509988) and were compared with the 16S rRNA sequences in NCBI database using BlastN to find the closest relatives.

### 2.5. Microbial Community Analysis and Bioinformatics

After 96 h of incubation, soil samples from five replicates of each plant species from the same batch were pooled. DNA was extracted from the soil samples using the NucleoSpin Soil kit (Macherey-Nagel, Düren, Germany) according to the manufacturer’s instructions. Bacterial 16S rRNA gene amplification was carried out on the V3 and V4 region of the 16S rRNA gene, with the forward primer: 341f (5′-CCTACGGGNGGCWGCAG-3′) and the reverse primer: 785r (5′-GACTACHVGGGTATCTAATCC-3′) [28], each primer was flanked by Illumina-specific overhang sequences, as described in the Illumina handbook [29]. The amplicon libraries were sequenced on an Illumina MiSeq platform, using MiSeq Reagent Kit v3 (600 cycle) and 20% PhiX control standard. Raw data obtained were deposited to the NCBI database under the accession number PRJNA694775.

Raw reads were analysed as described by Dumbrell et al. [30]. In brief, the sequence reads were trimmed using Sickle [31], error corrected with SPAdes [32] using the BayesHammer algorithm [33] and pair-end aligned with PANDASeq [34] using PEAR [35]. After dereplication, sequences were clustered into operational taxonomic units (OTUs) using VSEARCH [36] at the 97% identity level. OTUs with one read were removed, along with all chimeric sequences using reference-based chimera checking within UCHIME [37], then OTUs were taxonomically assigned using the RDP classifier [38]. Sequence data were rarefied to 19,359 sequences.

#### 2.5.1. Differences in Bacterial Community Composition

The OTU abundance table was used to test for the differences in bacterial community composition among the soil samples associated with the different types of plant species, isoprene concentrations and seasons, using PAST software version 2.17c [39]. The differences in community composition were calculated using NPMANOVA (*p*-value based on 9999 permutations). Patterns of community variation were visualised by non-metric multidimensional scale (NMDS) ordination of Bray–Curtis distance.

STAMP software [40] was used to identify OTUs and specified genera that significantly changed in relative abundance with the treatments. Analysis was performed as described by Aslam et al. [41] with default parameters, except that the parameters for filtering out were: *p*-value > 0.05; difference between proportions <0.2 or difference between ratios <1.5.

#### 2.5.2. Phylogenetic Analysis

Phylogenetic analysis was performed using an alignment of partial 16S rRNA gene sequences (constructed using MEGA-X) [42] from representatives of OTUs that significantly increase in relative abundance in the isoprene treatments and isoprene-degrading isolates, together with their closest relatives. The Neighbor-Joining tree [43] was constructed using the Maximum Composite Likelihood protocol [44] with 1000 bootstrap replicates [45].

### 2.6. Measurement of Isoprene-Degrading Activity of Bacterial Isolates

In order to confirm the degradation activity of isoprene by bacteria isolated from soil samples, each culture was inoculated into 9 mL of three different types of media, contained in 125 mL crimp-top serum vials sealed with PTFE/silicone septa. The media used included minimal medium broth (as described above) without isoprene, minimal medium broth with isoprene, and glucose/yeast broth (contained per 1 L of Milli-Q water: 20 g of glucose (Merck, Darmstadt, Germany) and 10 g of yeast extract (Difco, Oxford, UK) with isoprene. Isoprene was added to 7.2 × 10^5^ ppbv. Every three days, residual isoprene was measured by taking 100 µL of headspace gas with a glass gas-tight syringe and injecting into a Unicam 610 series GC-FID equipped with a 10% Apiezon L CW column, with the injector and detector temperature of 160 °C and the column temperature of 100 °C. Optical density (OD at 600 nm) was also measured every three days using a Jenway 7300 spectrophotometer. After 18 days of incubation, the cultures were subjected to serial dilution, spread on minimal medium agar and incubated with isoprene to confirm purity.

## 3. Results and Discussion

### 3.1. Isoprene Degradation by Soil Microbes Associated with Economic Crop Species and Framework Forest Tree Species

Soil samples beneath three framework forest tree species (council tree, wild oak and wild Himalayan cherry) and economic crop species (oil palm, rubber tree and sugar cane) (Table 1), collected at different seasons, were mixed with minimal medium and incubated with isoprene at two different concentrations. The percentage losses of isoprene over 96 h were between 4.4% and 30.9% for the soil samples incubated with isoprene at the lower concentration, and between 15.4% and 51.6% for samples incubated with isoprene at the higher concentration (Figure 1). There was a small loss of isoprene from the sterile controls; however, 31 out of the 36 treatments had significantly more isoprene loss than the corresponding sterile controls (Figure 1). The exceptions were rubber-tree soils and sugar-cane soils collected during rainy and winter seasons, incubated with the lower concentration of isoprene, and rubber-tree soils collected during summer season, incubated with the higher concentration of isoprene (Figure 1). After combining seasonal and concentration data, only rubber-tree soil slurries had significantly lower isoprene loss compared with the other five tree species (Anova Tukey HSD test; *p* < 0.05). These findings reveal that soil associated with various types of trees had the capacity for isoprene consumption, which suggests widespread distribution of isoprene degraders in tropical soils, as seen for other soil types [13,14,19,20]. There is no obvious reason for the differential capacity for potential isoprene degradation with the the rubber plantation soil, as rubber is a polymer of isoprene, and some bacteria, e.g., *Gordonia* spp., can degrade both [23]. From previous studies, there is no evidence of a relationship between plant isoprene production and isoprene consumption in associated soils, probably because soil microbiota primarily consume isoprene produced in the soil, e.g., by bacteria and fungi rather than from the atmosphere [15,16]. Isoprene production from the studied plants has only been documented for oil palm and rubber trees [10,11,12].

The percentage loss of isoprene from soils treated with the higher isoprene concentration was greater than that with the lower concentration (Anova Tukey HSD test; *p* < 0.05), which is consistent with data from temperate soils [14,19], although these studies applied much lower isoprene concentrations than our study.

Degradation in soils sampled in the summer was significantly lower than in soils collected in winter or the rainy season (Anova Tukey HSD test; *p* < 0.05), but there was no statistically different degradation between soils collected in the rainy season and winter. In the tropical climates of mountainous upper Northern Thailand and the inland plain of lower Northern Thailand, the main seasonally different factor in soils is moisture. Soil moisture in the summer season was lower than in winter and in the rainy season, whereas there was little difference in soil temperatures across the seasons (Appendix A). While our data on potential isoprene degradation are consistent with studies in which soils were incubated with different moisture content [13], Pegoraro et al. [46] observed that the optimal isoprene consumption rate was fully restored within 2 h of rewetting the soil. Overall, our study is indicative of the capacity for isoprene degradation in tropical soils, but further analyses are needed, especially using in-situ chambers at atmospheric isoprene concentrations, to understand fully how soil parameters and plant species influence the capacity for isoprene degradation.

### 3.2. Bacterial Community Analysis and Identification of Taxa That Increase in Relative Abundance after Isoprene Enrichment

NPMANOVA analysis revealed that bacterial communities derived from soils beneath framework forest trees were not statistically different from each other (F_2,26_ = 6.142, *p* = 1) (shown by the cluster on the left-hand side of the ordination plot, Figure 2a, with communities from wild Himalayan cherry, council tree and wild oak). By contrast, bacterial communities derived from the soils in economic crop plantations (oil palm, rubber and sugar cane) significantly differed from each other and from the framework forest tree group (F_1,50_ = 20.64, *p* < 0.01; Figure 2b). However, it should be noted that the soils associated with these two plant categories (framework trees or economic crops) were collected from different geographical locations, where altitude and climate also differed. The framework forest trees are found in a mountainous conservation area, i.e., undisturbed forest restoration area in the upper northern region of Thailand, whereas the economic crops were grown in an area of intensive agriculture in the plain of the lower northern region. Moreover, the soils associated with the framework forest trees were close to each other (within 500 m), while the crop plantations were approximately 1 km from each other. All of these factors (dominant plant species, diversity of surrounding plant species, altitude, geographic separation) may have influenced the soil microbiota and how it developed when incubated with isoprene [47,48,49].

Enrichment with isoprene led to a convergence in, and a significantly different, bacterial community composition compared with the soils that were not incubated with isoprene (pre-enrichment treatments) (F_1,50_ = 3.746, *p* < 0.05; Figure 2d). However, there was no overall significant difference between bacterial community compositions exposed to different isoprene concentrations (F_1,50_ = 0.8377, *p* = 1; Figure 2d). Although season had an influence on isoprene degradation potential, it had no significant effect on the bacterial community composition (F_2,50_ = 1.66, *p* = 0.052; Figure 2c). Given that seasonal differences in the microbiota are commonly observed in soils [50], including tropical rubber-plantation soils [51], it is probable that any seasonal difference was masked by the overriding selective pressure of enrichment with isoprene.

Prior to this study, little was known about isoprene-degrading microorganisms in tropical soils, with just a few studies analysing isoprene degraders in soil associated with oil palm trees, but growing in temperate environments [52,53], or in soil transported to the UK [20]. Oil-palm soil was found to have the highest abundance of isoprene monooxygenase genes (*isoA*), determined by quantitative PCR [20,54]. In these studies, genera such as *Rhodococcus, Gordonia, Novosphingobium, Pelomonas, Rhodoblastus, Sphingomonas* and *Zoogloea* were identified as isoprene-degrading bacteria by DNA stable isotope probing [20,52,53,55]. A wider variety of genera has also been implicated in isoprene degradation in temperate soils [17,19,20,54,55].

Our study had the advantage of investigating freshly sampled soils from the tropics, and analysing the changes in bacterial community composition shortly after isoprene degradation had started. However, the isoprene-degrading ability of genera that increased in relative abundance only provided evidence for a potential role in isoprene degradation. Nevertheless, our cultivation studies (see later) provided firm links between taxonomy and isoprene degradation. Several genera of bacteria increased significantly in relative abundance in soil enriched with isoprene (Figure 3), most notably *Acinetobacter, Comamonas, Pseudomonas* and *Rhodococcus*. Some genera, such as *Comamonas*, increased in soil from both categories of plants and at both concentrations of isoprene. *Pseudomonas*, however, increased with the lower concentration of isoprene, while *Rhodococcus* increased only at the higher isoprene concentration (Figure 3A).

Investigating bacterial community composition at the OTU level revealed patterns that were not seen at the genus level. For example, *Bacillus* OTU1153 (having 99.5% 16S rRNA similarity to *Bacillus luciferensis* strain LMG 18422) increased significantly in relative abundance in soils with isoprene (Figure 3B and Figure 4). The most abundant OTU from *Acinetobacter* (OTU6; Figure 4) showed variability between replicates and was significantly more abundant only when incubated at a lower concentration of isoprene (Figure 3B). *Rhodococcus* OTU11 (100% identity with *Rhodococcus pedocola* strain UC12) was much more abundant in the isoprene-enriched soils from framework forest trees than from economic crops, whereas *Pseudomonas* OTU27 (100% identity with *Pseudomonas multiresinivorans strain* ATCC 700690) had the opposite pattern (Figure 3B and Figure 4).

Many of these genera have typically been found to increase in isoprene-enriched soils or have been recognised as isoprene degraders, such as *Rhodococcus*, *Comamonas* and *Pseudomonas* [17,20,21,54,56]. In contrast, *Acinetobacter* has not been shown to have an isoprene-degrading capacity, although it is a widespread genus with broad catabolic capabilities, including the capacity to grow on short-chain alkanes [57] and alkenes [58]. OTU6, which became most abundant after isoprene enrichment, was identical in 16S rRNA gene sequence to the diesel-degrading *Acinetobacter oleivorans* strain DR1 isolated from a rice paddy field in South Korea [59]. Evidence from methods such as stable-isotope probing or cultivation is needed to confirm whether *Acinetobacter* species can grow with isoprene as the sole carbon and energy source. Similarly, significant isoprene-induced increases in the relative abundance of the common soil bacterial genus *Edaphobacter*, one of the most abundant Acidobacteria in soil, which plays an important role in forestry and agriculture [60], merit further investigation. These results are indicative of potentially novel isoprene-degrading taxa.

### 3.3. Isoprene-Degrading Bacteria Isolated from Soil Samples

Bacterial isolates were obtained from isoprene-enriched soils associated with wild Himalayan cherry, wild oak, sugar cane and oil palm. Eight isolates were cultured in media supplemented with isoprene, and all could use isoprene as the sole source of carbon and energy, demonstrated by simultaneous growth (compared with bacteria incubated in minimal medium without isoprene (Anova; *p* < 0.05)), and isoprene degradation (compared with uninoculated controls (Anova; *p* < 0.01)) (Figure 5). They grew to a far higher cell density when supplied with other carbon and energy sources in the form of glucose and yeast extract, but still degraded isoprene at similar rates to when isoprene was the only source of carbon and energy (except for the *Ochrobactrum* strain 17f; Figure 5).

Analysis of 16S rRNA gene sequences revealed that the isolates belong to various genera (Figure 4; Appendix A), some of which are known to have species that can degrade isoprene, i.e., *Arthrobacter*, *Bacillus* and *Klebsiella* [16]. Other isolates are the first of their genus to be described as isoprene degraders, i.e., *Ochrobactrum* (belonging to the Alphaproteobacteria) and *Friedmanniella, Isoptericola* and *Cellulosimicrobium* (all belonging to the Actinobacteria) (Figure 4). Seven isolates had >98.7% 16S rRNA gene sequence similarities to the type strains within the respective genera, but strain 39f had only 95.8% identity to *Friedmanniella spumicola* strain Ben 107 (Appendix A). Figure 4 also highlights that the isoprene-degrading isolates were mostly from different genera to those whose increase in relative abundance was determined by cultivation-independent methods, with the exception of Bacillus.

## 4. Conclusions

In this study, the isoprene-degrading ability of soil bacteria associated with tropical plant species, which are important in forestry and agriculture, was investigated. Microbial degradation of isoprene was found in soils associated with all plant species examined. These results suggest a widespread potential for isoprene degradation in tropical soils. Eight bacterial isolates obtained from the soil samples associated with both crop plants and forest framework trees were able to use isoprene as the sole source of carbon and energy. This provides a number of novel isoprene-degrading strains associated with tropical plant species with which to study the mechanisms and regulation of isoprene degradation by bacteria and determine their potential for cycling isoprene in tropical environments.

## Figures and Tables

**Figure 1 microorganisms-09-01024-f001:**
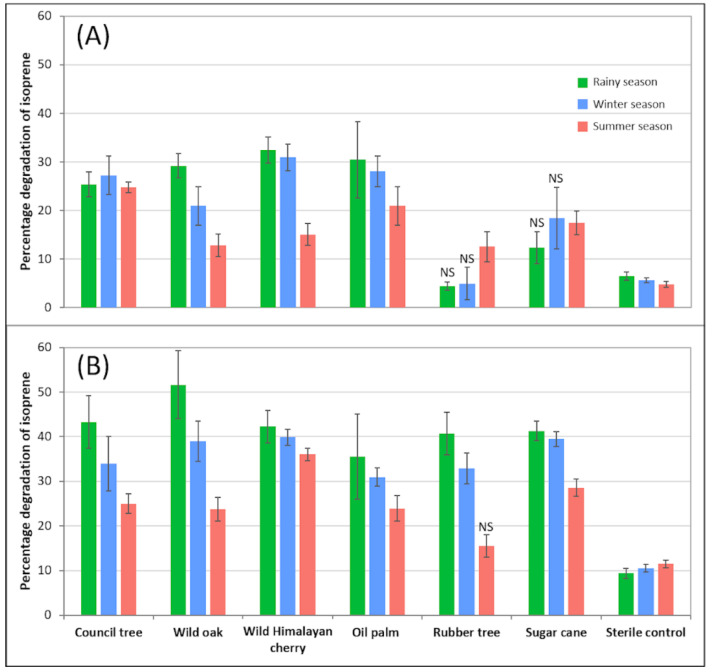
Percentage loss of isoprene over 96 h in microcosms containing 1 g of soil, incubated with isoprene at two different concentrations: (**A**) 7.2 × 10^5^ ppbv and (**B**) 7.2 × 10^6^ ppbv. *n* = 5. Error bars show ±SE. NS = the percentage loss was not statistically different from the sterile control (Anova; *p* < 0.05).

**Figure 2 microorganisms-09-01024-f002:**
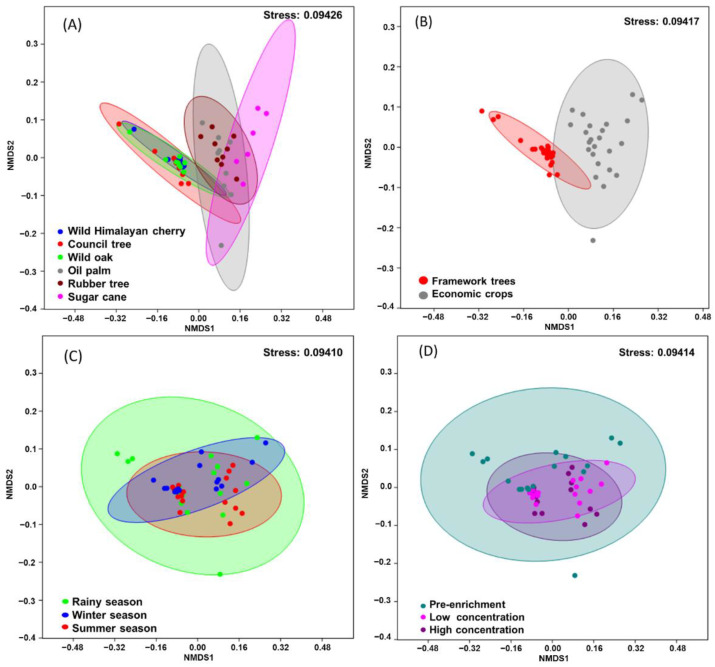
Non-metric multidimensional scaling (NMDS) analysis of bacterial communities, based on Bray–Curtis dissimilarity (using OTUs derived from 16S rRNA gene sequences). Each point represents a single community, with points closer together indicating compositionally similar communities. All panels show the same data with 95% confidence ellipses for (**A**) Plant species, (**B**) Type of plants (**C**) Seasons and (**D**) Isoprene concentrations.

**Figure 3 microorganisms-09-01024-f003:**
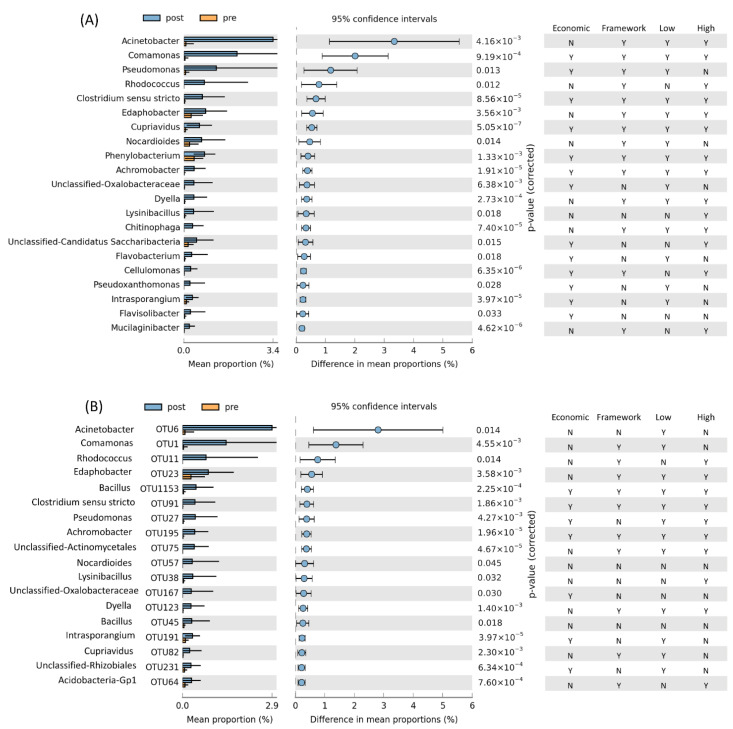
Taxa with significant increase in relative abundance in soils (based on all treatments combined) when incubated with isoprene (post) compared with before incubation (pre): (**A**) Genus level, (**B**) OTU level. The right-hand column shows whether the indicated genus or OTU was significantly more abundant (Y) or not (N) in soils from beneath Economic crops and Framework trees, and with lower isoprene concentration and higher isoprene concentration, each relative to their respective soils before incubation. Analysis was performed using STAMP software. See Methods for details.

**Figure 4 microorganisms-09-01024-f004:**
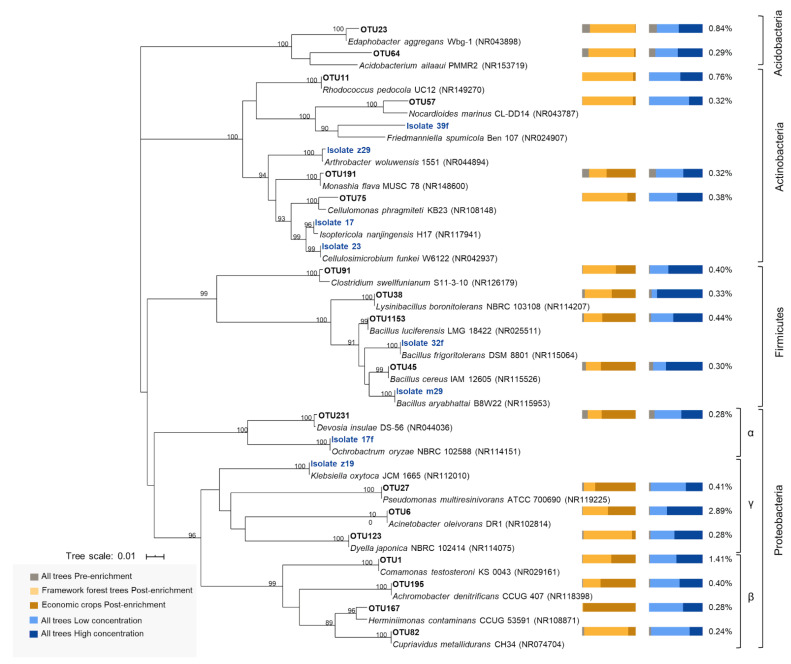
Neighbor-joining phylogenetic analysis of 16S rRNA genes from 18 OTUs showing an increase in abundance when grown with isoprene and eight isoprene-degrading isolates (in blue), aligned with their closest relatives. This analysis involved 402 positions, sum of branch length = 1.66. Bootstrap values greater than 80% are displayed. Bar charts on the right show the relative abundance of OTUs for the indicated treatments (pre-enrichment: soil samples before incubation with isoprene, post enrichment: soil samples after incubation with isoprene). Percentage values on the right show the mean relative abundance of OTUs in all post isoprene enrichments.

**Figure 5 microorganisms-09-01024-f005:**
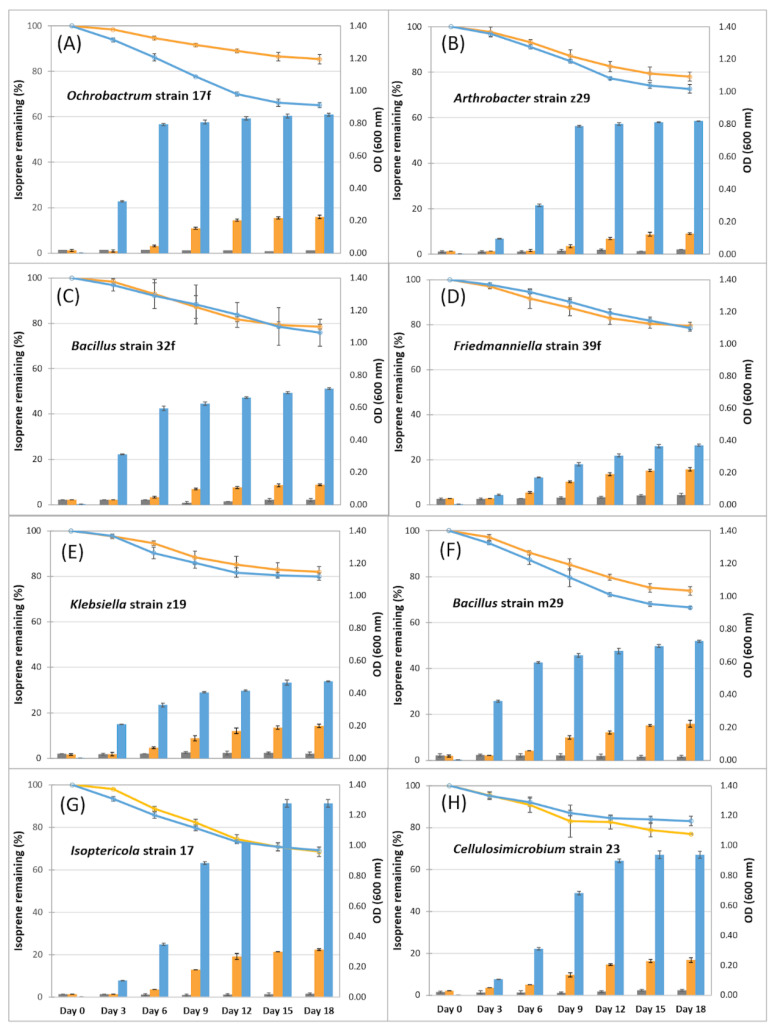
Growth and isoprene degradation by bacteria isolated from soil samples (**A**–**H** show data from eight different isolates). The growth of isolates, measured by OD_600_, is shown for: minimal medium only (grey bars, scale exaggerated 10-fold), minimal medium with isoprene (orange bars, scale exaggerated 10-fold) and glucose/yeast medium with isoprene (blue bars). Isoprene was supplied at 7.2 × 10^5^ ppbv). Degradation is shown by the percentage of isoprene remaining in cultures grown in minimal medium supplied with isoprene (orange lines) and glucose/yeast medium with isoprene (blue lines). *n* = 3. Error bars show ±SD.

**Table 1 microorganisms-09-01024-t001:** Plant species and associated soils, selected for this study.

Type of Plant	Common Name	Scientific Name
Framework tree	Council tree	*Ficus altissima* BI.
	Wild oak	*Quercus semiserrata* Roxb.
	Wild Himalayan cherry	*Prunus cerasoides* D. Don
Economic crop	Oil palm	*Elaeis guineensis* Jacq.
	Rubber tree	*Hevea brasiliensis* Muell. Arg.
	Sugar cane	*Saccharum officinarum* L.

## Data Availability

Publicly available datasets were analyzed in this study. This data can be found in the NCBI database under the accession number PRJNA694775 and MW509981–MW509988.

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
