# Peer review of "Isoprene-Degrading Bacteria from Soils Associated with Tropical Economic Crops and Framework Forest Trees"

_microorganisms, 2021, doi:10.3390/microorganisms9051024_

Round 1
Reviewer 1 Report
This work provides a number of isoprene-degrading strains associated with tropical plant species whose further characterisation will help to understand isoprene biodegradation and gene regulation mechanisms. The work is interesting and well presented, however some clarifications regarding the methodology need to be included.
It would be interesting for a better understanding to include in materials and methods a better description of how the enrichment cultures are obtained from the collection of the soil samples.
Line 203 : …were mixed with minimal medium and incubated with isoprene at two different concentrations. Please specify and also include in materials and methods, whether isoprene is the carbon source included in this minimal medium or a different one.
Lines 267-268: Enrichment with isoprene led to a convergence in, and a significantly different, bacterial community composition compared with the pre-enrichment treatments. Please clarify what these pre-enrichment treatments are and also the differences with the post-enrichments in figure 4.
Author Response
Reviewer 1
This work provides a number of isoprene-degrading strains associated with tropical plant species whose further characterisation will help to understand isoprene biodegradation and gene regulation mechanisms. The work is interesting and well presented, however some clarifications regarding the methodology need to be included.
It would be interesting for a better understanding to include in materials and methods a better description of how the enrichment cultures are obtained from the collection of the soil samples.
Line 203: …were mixed with minimal medium and incubated with isoprene at two different concentrations. Please specify and also include in materials and methods, whether isoprene is the carbon source included in this minimal medium or a different one.
We agree with Reviewer 1 that more detail is required. Therefore, we have now added full details of the medium and specified that isoprene is the sole carbon and energy source (line 105-113). As a consequence the reference to the PhD thesis by Murphy et al. [22] has been changed to an original reference by Fahy et al. (2006).
Lines 267-268: Enrichment with isoprene led to a convergence in, and a significantly different, bacterial community composition compared with the pre-enrichment treatments. Please clarify what these pre-enrichment treatments are and also the differences with the post-enrichments in figure 4.
This information has now been added in two places (lines 280 and the legend to Figure 4).
Reviewer 2 Report
L34: „novel bacteria” – very general; in what sense 'novel'? It is worth clarifying or writing differently.
L79-82: Very colloquial vocabulary e.g. links, consuming...
L85: what does "asepticaly collected" mean? it means how? Exactly.
L85-93: how many samples were taken (g)? How many replicates for each combination? Any info on these soils? What type, pH etc.?
Subchapters 2.6 and 2.7 can actually be subsumed under 2.5, so I suggest dividing them into 2.5.1 and 2.5.2.
L219: Gordonia spp. should be written in italics.
L347-357: names of species and genera are written in italics.
Figure 5: and any statistical test for these results? Are they statistically significantly different?
References: Of the 57 papers, 19 (33.3%) are below 2005 with as many as 12 (21%) below 2000. A bit too many of these old citations.
Author Response
Reviewer 2
L34: „novel bacteria” – very general; in what sense 'novel'? It is worth clarifying or writing differently.
We agree that the original text was not clear, and so we have changed it as follows: “Eight isoprene-degrading bacterial strains were isolated from the soils, and among these, four belong to the genera Ochrobactrum, Friedmanniella, Isoptericola and Cellulosimicrobium, which have not been shown previously to degrade isoprene.” Please note that this adjustment required removal of some text from the preceding sentence to maintain the Abstract’s word count below 200. The deleted text “their isoprene consumption varied between 14 and 34%” is actually not so meaningful without further details. We thank the reviewer for helping us to produce a clearer abstract.
L79-82: Very colloquial vocabulary e.g. links, consuming...
We agree that this was written in a rather casual way, and have changed the text to make the aims much more clear and concise as follows: “Our aims were: 1) to determine the potential for isoprene degradation in tropical soils associated with different plants that are important in agriculture and forestry; 2) to identify changes in bacterial community composition during isoprene degradation, and 3) to isolate and characterise soil bacteria responsible for isoprene degradation.” (new lines 81-84)
L85: what does "asepticaly collected" mean? it means how? Exactly.
Details have been added (new line 80).
L85-93: how many samples were taken (g)? How many replicates for each combination? Any info on these soils? What type, pH etc.?
The mass of sample taken has been added in line 88, and we already provided the detail about five replicates in line 96 and in relevant figure legends. Some information about the soils was already in the supplementary data (Figure S1, S2), but further information has been added, including the pH as requested in a new Supplementary Table (Table S1). As a consequence, we have removed Figure S1 and S2, and the old Table S1 has been changed to Table S2.
Subchapters 2.6 and 2.7 can actually be subsumed under 2.5, so I suggest dividing them into 2.5.1 and 2.5.2.
Changed as requested.
L219: Gordonia spp. should be written in italics.
”Gordonia” was in italics, “spp.” should not be in italics.
L347-357: names of species and genera are written in italics.
These are already in italics.
Figure 5: and any statistical test for these results? Are they statistically significantly different?
The information indicating that these results are statistically significant has been added to line 365-366.
References: Of the 57 papers, 19 (33.3%) are below 2005 with as many as 12 (21%) below 2000. A bit too many of these old citations.
It is proper to cite landmark papers (like those from Cleveland and Yavitt, 1997; 1998), and when citing the methods, such as neighbor joining phylogenetic inference, it is correct to cite the original papers (i.e. Saitou and Nei, 1987). In fact, most of the pre-2000 citations are associated with classic methods. We are up to date in the field (11 citations are post 2018) and have given due credit to older work. We hope the reviewer accepts that no changes are required here.